# Curvature Analysis of Seed Silhouettes in *Silene* L.

**DOI:** 10.3390/plants12132439

**Published:** 2023-06-25

**Authors:** Emilio Cervantes, José Luis Rodríguez-Lorenzo, José Javier Martín-Gómez, Ángel Tocino

**Affiliations:** 1Instituto de Recursos Naturales y Agrobiología (Consejo Superior de Investigaciones Científicas), Cordel de Merinas 40, 37008 Salamanca, Spain; jjavier.martin@irnasa.csic.es; 2Plant Developmental Genetics, Institute of Biophysics v.v.i, Academy of Sciences of the Czech Republic, Královopolská 135, 612 65 Brno, Czech Republic; rodriguez@ibp.cz; 3Departamento de Matemáticas, Facultad de Ciencias, Universidad de Salamanca, Plaza de la Merced 1-4, 37008 Salamanca, Spain; bacon@usal.es

**Keywords:** Bézier curve, complexity, curvature, development, Elliptic Fourier Transform, models, morphology, seed, systematics

## Abstract

The application of seed morphology to descriptive systematics requires methods for shape analysis and quantification. The complexity of lateral and dorsal views of seeds of *Silene* species is investigated here by the application of the Elliptic Fourier Transform (EFT) to representative seeds of four morphological types: smooth, rugose, echinate and papillose. The silhouettes of seed images in the lateral and dorsal views are converted to trigonometric functions, whose graphical representations reproduce them with different levels of accuracy depending on the number of harmonics. A general definition of seed shape in *Silene* species is obtained by equations based on 40 points and 20 harmonics, while the detailed representation of individual tubercles in each seed image requires between 100 and 200 points and 60–80 harmonics depending on their number and complexity. Smooth-type seeds are accurately represented with a low number of harmonics, while rugose, echinate and papillose seeds require a higher number. Fourier equations provide information about tubercle number and distribution and allow the analysis of curvature. Further estimation of curvature values in individual tubercles reveals differences between seeds, with higher values of curvature in *S. latifolia*, representative of echinate seeds, and lower in *S. chlorifolia* with rugose seeds.

## 1. Introduction

The Caryophyllaceae Juss. comprises ca. 100 genera and 3000 species of herbs and small shrubs [1,2] of a cosmopolitan distribution and characterized by a peripheral position of the embryo in the developing seed [3], anatropous to campylotropous ovules [4], with an interesting diversity in seed shape [5,6,7,8,9,10,11,12,13,14].

Morphological analysis of seeds in the Caryophyllaceae focuses on two aspects: overall seed shape and detailed seed surface structure. Cardioid-derived models have been applied to the quantification of overall shape in lateral views of the seed [5,6,7], while ellipse-based models fit well to dorsal views of the seed in many species [8]. The application of models to seed shape quantification contributes to the identification of useful characters for taxonomy. Seeds of *Silene* L. subg. *Behenantha* conform better to a cardioid than those of *Silene* subg. *Silene* [5]. Seed images of species of sect. *Melandrium* fit better to models derived from a modified cardioid closed in the hilum region, whereas seeds of sect. *Silene* fit better to open models [9].

In relation to the seed surface, and based on their geometric properties, *Silene* seeds were divided into four groups: smooth, rugose, echinate and papillose [10,11]. Smooth seeds are defined by the absence of visible protuberances and this type was already described by other authors working with *Silene* [12,13,14] or related genera, such as *Arenaria* L. [15,16,17], *Minuartia* L. [18] and *Moehringia* L. [19]. In *Silene*, species with smooth seeds belong to *S.* subg. *Silene* sec. *Silene* such as *S. apetala* Willd., *S. borderei* Jord., *S. colorata* Poir., *S. damascena* Boiss. and Gaill., *S. diversifolia* Ott, *S. legionensis* Lag., *S. micropetala* Lag., *S. nicaeensis* All., *S. pomelii* Batt. subsp. *adusta* (Ball) Maire, *S. secundiflora* Ott, *S. vivianii* Steud.), other sections in *S.* subg. *Silene* such as *S. crassipes* Fenzl (sec. *Lasiocalycinae*), *S. colpophylla* Wrigley and *S. ramosissima* Desf. (sec. *Siphonomorpha*), and more rarely to *S.* subg. *Behenantha,* such as *S. baccifera* Roth (sec. *Cuccubalus*) and *S. littorea* Brot. (Sec. *Psammophilae*) [10,11].

Smooth seeds are characterized by high values of circularity and solidity in their lateral views, while, in dorsal views, higher values of circularity are shared by echinate and rugose seeds [10]. Papillose seeds are characterized by the lowest values of circularity and solidity in both lateral and dorsal views. Species included in this group are *S. holzmani* Heldr. ex Boiss. (sec. *Behenantha*), *S. laciniata* Cav. (sec. *Physolychnis*), *S. magellanica* (Desr.) Bocquet (sec. *Physolychnis*) and *S. perlmanii* W.L.Wagner, D.R.Herbst and Sohmer (sec. *Sclerophyllae*) [10].

Due to the large number of species, as well as infraspecific variation, it is important to quantitatively define seed surface structure properties and tubercle curvature in species of *Silene* and other genera in the Caryophyllaceae. The application of the Elliptic Fourier Transform (EFT) to seed images can provide new methods and models for seed shape description and quantification [20,21,22]. Once the seed silhouettes are represented by elemental functions, it is possible to calculate the corresponding curvature values. Curvature of a plane curve is a descriptive measure of shape that measures the rate at which the tangent line turns per unit distance moved along the curve. Departing from Bézier curves representing the root silhouettes, curvature was measured in *Arabidopsis* Heinh in Hall and Heinh (Brassicaceae) roots showing reduced values in ethylene-insensitive mutants (*etr1-1* and *ein2-1*) [23], as well as under hydrogen peroxide treatment [24]. In addition, curvature analysis allowed researchers to differentiate morphotypes in wheat kernels [25] and to define three groups of seeds in cultivated grapevine (*Vitis vinifera* L., Vitaceae) [26].

Closed plane curves based on EFT reproduced the seed silhouettes of representative *Silene* species [20]. The number of harmonics required to obtain curves reproducing the silhouettes provides an idea of the complexity of the seed surface and can be related to the four described types [10,11]. Curvature analysis based on EFT curves provides information about the geometry of the tubercles [20]. The analysis of seed surface structure has been applied to seeds of four *Silene* species: *S. colorata* Poir., *S. chlorifolia* Sm., *S. latifolia* Poir. and *S. perlmanii* W.L.Wagner, D.R.Herbst and Sohmer, representative of smooth, rugose, echinate and papillose seeds, respectively [10,11]. First, EFT curves are described for the lateral and dorsal views of seeds, and curvature analysis is performed on the EFT curves. Curvature analysis based on Bézier curves is also applied to the individual tubercles in the seeds of *S. chlorifolia* and *S. latifolia* providing an example for comparison of tubercle shape between seeds, populations or species. The analysis of curvature based on the combination of both EFT and Bézier curves provides a solid basis for the description and comparison of seed surface structure in *Silene*.

## 2. Results

### 2.1. General Morphological Aspects of the Seeds

Table 1 contains a summary of the morphological characteristics for the lateral and dorsal views of the seeds used in this work. In the lateral view, the seeds of *S. perlmanii* had the smallest area and lowest values of circularity and solidity, as was reported for the papillose-type seeds [10,11]. In the dorsal view, the highest values of solidity corresponded to the seeds of *S chlorifolia* and *S. latifolia*, representing the groups of rugose and echinate seeds, respectively. The high values of solidity and relatively low coefficients of variation are indicative of relatively stable morphological conditions in the seed populations.

### 2.2. Elliptic Fourier Transform and Curvature: General Aspects

Closed curves resulting from EFT analysis of seed contours reproduced the lateral and dorsal views of seed silhouettes [20]. The EFT curves for eight images representing the lateral and dorsal views of *Silene* seeds are presented. Similarity between the curve and the image silhouette is recognized by the coincidence between seed surface and EFT curve. Once a similarity was reached, Fourier curves representing the seed images were the subject for curvature analysis (Method 1: curvature based on EFT [20]; see Appendix A). In the case of seeds with tubercles, these and the protuberances of the EFT curve coincide. Curvature values were estimated for each EFT curve—including all the tubercles in a single analysis, and, later, individually for representative tubercles (Method 2: Curvature on individual tubercles, based on Bézier curves [23,24,25,26]). Both methods of curvature measurement, based, respectively, on EFT and Bézier curves, were applied to seed images of the three tuberculate species (*S. chlorifolia*, *S. latifolia* and *S. perlmanii*), while Method 1 alone was enough to describe the surface of *S. colorata* seeds. Notice the qualitative difference between the two methods: a study of the global form provided by Method 1 *versus* a local analysis obtained from Method 2.

### 2.3. Smooth Seeds Are Represented by Curves with a Low Number of Harmonics

Figure 1 and Figure 2 show the original seed images and the EFT closed curves for the lateral and dorsal views of *S. colorata*, together with their corresponding curvature values along the curves. To obtain the EFT curves, 40 points were taken from the surface of the seed images, and the EFT images represented equations with 20 harmonics. In the lateral view, maximum curvature values correspond to the micropile region (Figure 1) and, in the dorsal view, to the two concave regions in the upper and lower sides of the seed image (Figure 2). The other peaks of curvature correspond to irregularities of the seed surface.

Lateral and dorsal views of *Silene colorata* seeds were accurately represented by 40-point-derived curves with 20 to 30 harmonics. In contrast, for tuberculate seeds, models derived from 40-point curves resemble the general shape, without considering the tubercles. These models are easy to develop and can be useful to obtain visual information on the general seed shape (Figure 3) for characteristics such as solidity and roundness (or aspect ratio). However, adjusting the surface contour to include the tubercles requires models obtained with more points.

### 2.4. Seeds with Tubercles Require a Higher Number of Harmonics

The curves resulting from the application of the EFT with 100–250 points and 40–80 harmonics to the lateral and dorsal views of *S. chlorifolia*, *S. latifolia* and *S. perlmanii* are shown together with their corresponding curvature values (Figure 4, Figure 5, Figure 6, Figure 7, Figure 8 and Figure 9).

In the lateral view of *S. chlorifolia* and *S. latifolia* seeds, there are less tubercles and with lower curvature values in the regions around the micropile. Curvature values in the tubercles comprise between −4 and 1 in *S. chlorifolia* and −8 and 5 in *S. latifolia*. Nevertheless, the highest values are due to small irregularities in the process of generation of the curve, and real tubercle curvature comprises between −2 and 1 *S. chlorifolia* and between −5 and 4 for *S. latifolia*. Tubercle number and absolute curvature values are higher in *S. latifolia*. In the seeds analyzed, in both species, the tubercles are of regular size and shape.

In the dorsal view of seeds from both species, the tubercles were concentrated in the seed poles, being more pronounced and with higher curvature values than in the lateral sides (Figure 6 and Figure 7).

Curves reproducing the silhouette of the lateral view of *S. perlmanii* were derived from the selection of 180–200 points in the application of EFT (Figure 8). The tubercles were larger than in *S. chlorifolia* and *S. latifolia* in relation to seed size. Estimated curvature values were between −3 and 1.

The curve representing the dorsal view of *S. perlmanii* reproduced 23 individual tubercles of curvature values between 1 and 35 (Figure 9).

### 2.5. Curvature Analysis on Individual Tubercles

Figure 10 shows the curvature analysis of individual tubercles, numbers 23, 1, 2 and 3, from the dorsal view of *S. perlmanii* represented in Figure 9. Curvature values are of 1.2 for tubercles 23, 1 and 2, and 2.1 for tubercle number 3. In contrast with *S. chlorifolia* and *S. latifolia*, the tubercles of *S. perlmanii* present great diversity in size and shape. 

The results of Fourier analysis indicated differences in curvature values between species. To investigate this in more detail, curvature analysis was performed in *S. chlorifolia* and *S. perlmanii*, two species that have regular tubercles. Figure 11 shows representative samples of a tubercle of each species with their Bézier curve and corresponding curvature analysis. Table 2 presents the comparison of means (ANOVA) for maximum and average curvature values in six tubercles of each species. The differences were significant (*p* < 0.05) both for maximum and mean values in the comparison between *S. chlorifolia* and *S. perlmanii*.

The curvature was also measured in six tubercles of each of three seeds (eighteen tubercles total) of *S. chlorifolia* and *S. latifolia* and the mean values were compared. The results are shown in Table 3.

## 3. Discussion

The field of descriptive systematics aims at discovering the patterns in nature and how they vary between organisms, populations and species [27]. This requires the application of mathematical protocols to define, quantify and compare the shapes in the organisms [28]. 

The species of the genus *Silene* L. have a remarkable variation in geographical distribution, breeding systems and ecological relationships. Due to their short life-cycles, facility to breed and the growing availability of genetic resources, they can be considered as models for ecology and evolution [29]. In addition, *S. latifolia* has heteromorphic sex-determination with an evolving non-recombining y region rich in repetitive DNA that provides a unique system for the study of the origin and modification of sex chromosomes [30]. In addition, an interesting seed shape diversity makes *Silene* a useful model for studying variations in seed morphology [5,6,7,8,9,10,11,12,13,14].

In many species of *Silene*, as well as in other species of Caryophyllaceae, seeds have tubercles arranged along the seed surface. Typically, 20 to 60 tubercles are observed in the lateral view and a smaller number in the dorsal view. Size and shape of the tubercles, as well as the regularity of their distribution, vary among species and among populations of the same species, making it possible to search for associations between tubercle characteristics and genetic or environmental factors. Infraspecific variation concerning tubercle size and shape has been reported in other genus of the Caryophyllaceae, such as *Arenaria* L., *Acanthophyllum* L. as well as *Silene* [14,16,31], and the variations in tubercle shape have been attributed to geographical and ecological factors or, in contrast, to taxonomic differences [16]. The study of seed surface variation will benefit from new quantitative methods for the description of tubercle morphology and seed surface.

The observation by optical microscopy of seeds of 100 species of *Silene* allowed for their classification into four groups according to their silhouettes: smooth, rugose, echinate and papillose [10,11]. Accurate representation of the seed surface structure was obtained by the application of Fourier Transform to seed images [20]. Subsequently, in this article, we have investigated the differences between representative species of the four morphological types related to the representation of their seed silhouettes by EFT. The main geometric features of the silhouettes of smooth seeds are represented with EFT curves derived from 40 points selected in the seed surface and 20 harmonics. This result agrees with estimates of 10 harmonics for reproducing the shape of leaves [32] and makes EFT with low harmonic number an interesting tool for the representation of general aspects of seed shape. Nevertheless, Fourier analysis applied to the species of tuberculate seeds (rugose, echinate and papillose), required a higher number of harmonics to define well the individual protuberances. Thus, although Fourier analysis with a low number of harmonics can discriminate successfully between various seed morphotypes, only with a higher number of harmonics can the morphological properties, size, shape and distribution of the tubercles can be analyzed.

To obtain curves adjusting to the protuberances at least 100 points are required, and the accuracy increases with higher numbers up to 250 or even more. Equations of 60 to 80 harmonics are sufficient in most cases, but more may be necessary to have detailed representation of the tubercles.

In addition to curvature analysis on curves derived from EFT, the analysis was focused (involving higher precision) on individual tubercles. The comparison revealed lower curvature values in *S. chlorifolia* (rugose seeds) and higher in *S. latifolia* (echinate). The application of the method to diverse populations of these species is required to confirm that a range of curvature values is a property of each species. In addition, the application to different species of each of the groups (rugose, echinate and papillose) will tell whether curvature values may be associated with the general morphology of the tubercles. A constant curvature value observed in the seeds of *S. chlorifolia* is related to lower curvature values and the proximity between mean and maximum values.

The method presented here opens the way to the analysis of size, shape and distribution of tubercles along the seed surface. The reported results remark upon the difference among seed types based on cell surface, with smooth seeds being characterized by a profile represented by low number of points, and on the other side, papillose seeds with numerous large tubercles that can only be represented by EFT when a large number of points are considered. Both types are distinguished also by their extreme values of circularity (highest in smooth seeds, lowest in papillose seeds) [10]. In between these two types remain the other two groups, rugose and echinate, here represented by *S. chlorifolia* and *S. latifolia,* respectively. While the tubercles in both are distributed more regularly than in papillose seeds, the results show differences with increased curvature values in *S. latifolia*. The results with other species and populations will demonstrate whether this is a property of echinate seeds, in contrast with rugose seeds, or if these differences are due to the species or populations chosen.

## 4. Materials and Methods

### 4.1. Silene Seeds

The populations of seeds analyzed in this work are listed in Table 4.

### 4.2. Seed Images

For the analysis of individual tubercles, photographs were taken with a Nikon Stereomicroscope Model SMZ1500 (Nikon, Tokio, Japan) equipped with a 5.24 megapixel Nikon DS-Fi1 of camera (Nikon, Tokio, Japan); lateral and dorsal views used in FET analysis were taken with a Nikon Z6 camera (Nikon, Tokio, Japan), equipped with an objective AF-S Micro NIKKOR 60 mm f/2.8G ED (Nikon, Tokio, Japan).

### 4.3. Elliptic Fourier Transform (EFT)

The application of EFT to any closed plane figure results in a curve that mimics the silhouette of the original figure and is amenable to curvature analysis. For this, a series of points were selected at regular intervals on the seed silhouette (Figure 12). The function, whose graphic approximates the shape, is a combination of trigonometric expressions; then, its expression allows for calculating the curvature values along the curve [20]. The program to obtain an EFT curve from a series of points and the application to the four seed types was made available (see Appendix A). The points were taken starting from the right side of the seed image silhouette and moving clockwise. Different curves can result from the same image depending on the number and positions of the points taken, as well as the number of points selected in the curve construction process (number of harmonics). It is important to avoid the duplication of points and to take a similar number of points at equivalent distances in the different samples when a comparative analysis is sought.

The process was divided in two consecutive methods. First (Method 1), 40 to 50 points were taken at regular intervals from the seed silhouettes of all four types and 20 or 30 points were selected for the Fourier curves. Method 1 resolved the silhouettes for lateral and dorsal views of *S. colorata* (smooth seeds), as well as the overall shape for all four species. For the tuberculate species (*S. chlorifolia*, *S. latifolia* and *S. perlmanii*), more points were taken at regular intervals from the seed silhouettes and Fourier curves performed with 60 harmonics or more were needed to fit the curve and the seed silhouette (Method 2). The program to obtain an EFT curve also contains the code for the calculation of curvature values along the curve and their corresponding plot (Figure 12).

### 4.4. Curvature Analysis

Curvature values were calculated either from the EFT curves (whole-seed images) or for individual tubercles (See Appendix A). In the figures, the curvature values are represented in reverse sense to the direction of the curve (starting at the last point and moving counterclockwise; Figure 12). Curvature values below the horizontal axis belong to peaks pointing towards the center of the seed, while peaks with positive values correspond to protuberances. Curvature values were determined for individual tubercles of each species according to established procedures [23,24,25,26] (See Appendix A). In the measurements of curvature for individual tubercles, the points were taken either manually, as represented in Figure 11 and Figure 12, or automatically with the function Analyze line graph of Image J. In the first case (points taken manually; Figure 11 and Table 2), six tubercles were selected from the lateral views of representative seeds of *S. chlorifolia* and *S. latifolia*, and their maximum and mean curvature values determined. In the case of points taken automatically (Table 3), six tubercles of three representative seeds for each species were analyzed.

### 4.5. Statistical Analysis

ANOVA was used to show significant differences between populations for the measured variables. In the case of the comparison of morphological characters involving four populations, ANOVA was followed by Tukey test to provide specific information on which means were significantly different from one another. Statistical analyses (ANOVA) were carried out on IBM SPSS statistics v28 (SPSS 2021).

## 5. Conclusions

Fourier analysis has been applied to four representative morphotypes of *Silene* seeds based on seed surface structure: smooth, rugose, echinate and papillose. The method can successfully discriminate between various seed groups. The surface of smooth seeds, with no visible tubercles and higher circularity values, is represented by EFT equations with 40 harmonics. EFT opens the way to the analysis of seed surface protuberances. Curvature analysis applied to individual tubercles revealed differences between representative seeds of rugose and echinate groups, with lower values in rugose seeds and higher in tubercles of echinate seeds.

## Figures and Tables

**Figure 1 plants-12-02439-f001:**
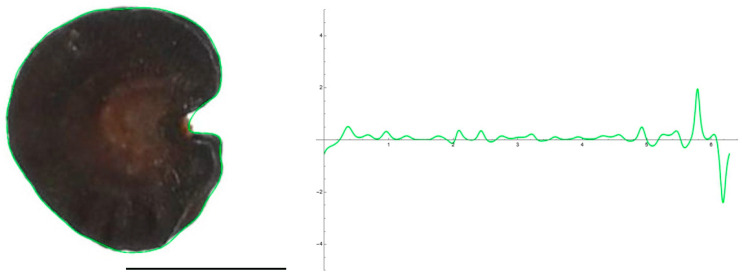
EFT and curvature analysis in a smooth seed. Lateral view of a seed of *S. colorata* with the EFT curve superimposed in green and the corresponding curvature values plotted. This is a smooth-type seed, lacking tubercles, and the maximum and minimum curvature values correspond to the micropilar region The EFT curve resulted from 43 points taken equidistantly along the seed silhouette and 20 harmonics, following the protocol published [20]. The program that provides the EFT curve contains the algorithm to plot the curvature values along it (see Appendix A). Bar represents 1 mm.

**Figure 2 plants-12-02439-f002:**
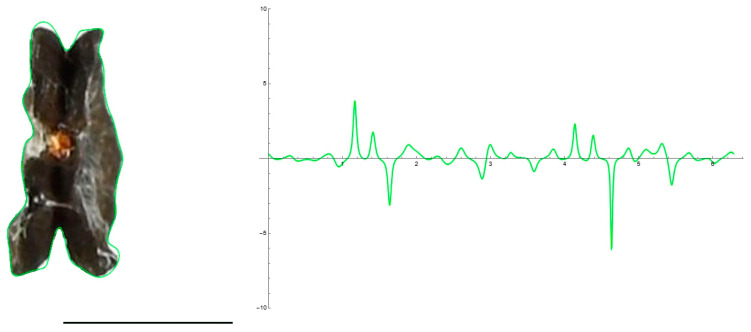
EFT and curvature analysis in a smooth seed. Dorsal view of a seed of *S. colorata* with the EFT curve superimposed in green and the corresponding curvature values plotted. This is a smooth-type seed, lacking tubercles and the maximum and minimum curvature values correspond to the upper and lower concavities. The EFT curve resulted from 38 points taken at regular distances along the silhouette and 20 harmonics, following the protocol published [20]. The program that provides the EFT curve contains the algorithm to obtain and represent the curvature values along it (see Appendix A). Bar represents 1 mm.

**Figure 3 plants-12-02439-f003:**
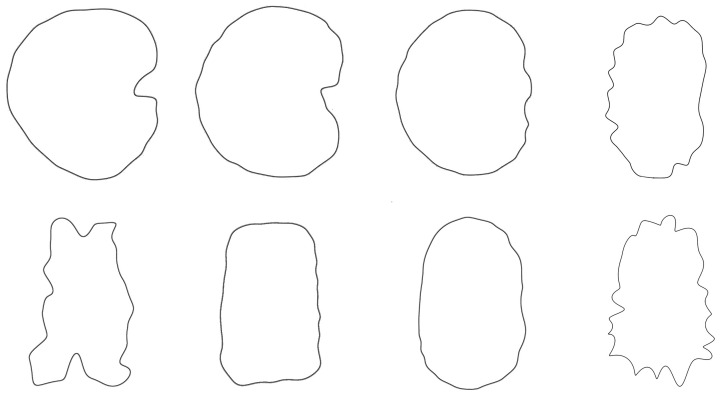
EFT curves with a low number of harmonics. EFT curves representing the silhouettes corresponding to seeds of *Silene colorata*, *S. chlorifolia*, *S. latifolia* and *S. perlmanii* (left to right). Above: lateral views. Below: dorsal views. Fourier analysis [20] was performed taking 40 points from the seed surface, and the resulting EFT curves obtained from 20 harmonics.

**Figure 4 plants-12-02439-f004:**
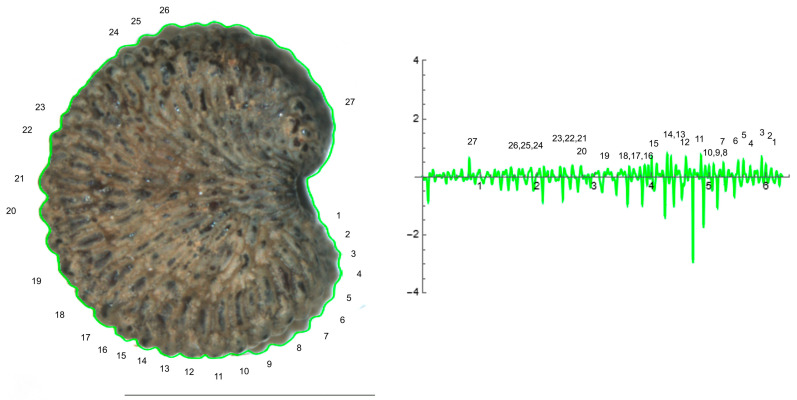
EFT and curvature analysis in a rugose seed. Lateral view of a seed of *S. colorata* with the EFT curve superimposed in green and the corresponding curvature values plotted. The EFT curve resulted from 193 points taken along the silhouette and 96 harmonics, following the protocol published [20]. The numbers indicate the correspondence between individual tubercles and their curvature values. The program that provides the EFT curve contains the algorithm to obtain and represent the curvature values along it (see Appendix A). Rounded tubercles are disposed regularly and the maximum and minimum curvature values correspond to individual tubercles. Bar represents 1 mm.

**Figure 5 plants-12-02439-f005:**
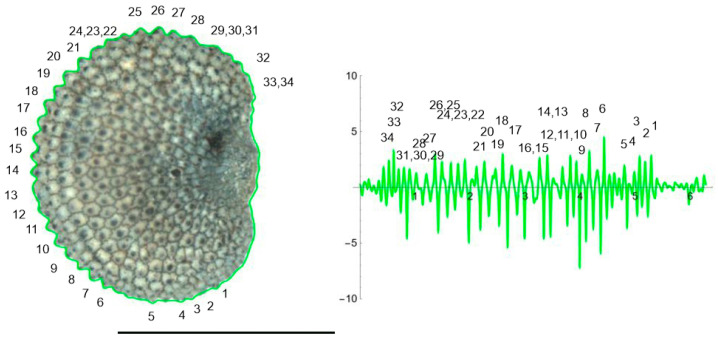
EFT and curvature analysis in an echinate-type seed. Lateral view of a seed of *S. latifolia* with the EFT curve superimposed in green and the corresponding curvature values plotted. The EFT curve resulted from 228 points taken along the silhouette and 74 harmonics, following the protocol published [20]. The numbers indicate the correspondence between individual tubercles and their curvature values. The program that provides the EFT curve contains the algorithm to obtain and represent the curvature values along it (see Appendix A). This is an echinate-type seed, with acute tubercles disposed regularly and the maximum and minimum curvature values correspond to individual tubercles. Bar represents 1 mm.

**Figure 6 plants-12-02439-f006:**
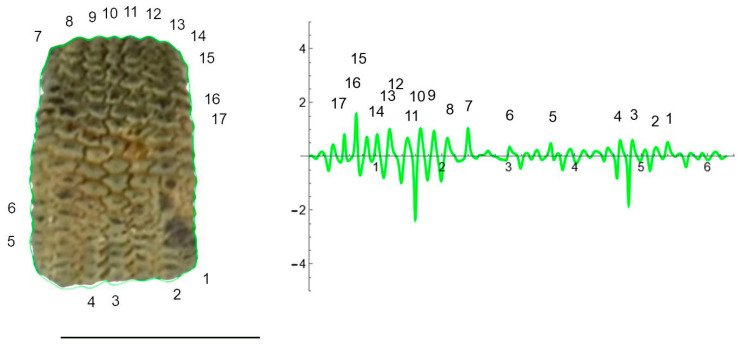
EFT and curvature analysis in a rugose seed. Dorsal view of a seed of with the EFT curve superimposed in green and the corresponding curvature values plotted. The EFT curve resulted from 102 points taken along the silhouette and 40 harmonics, following the protocol published [20]. The numbers indicate the correspondence between individual tubercles and their curvature values. The program that provides the EFT curve contains the algorithm to obtain and represent the curvature values along it (see Appendix A). Maximum curvature values correspond to individual tubercles in the poles. Bar represents 1 mm.

**Figure 7 plants-12-02439-f007:**
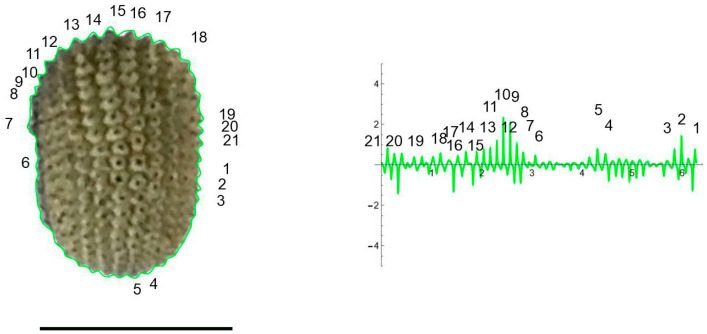
EFT and curvature analysis in an echinate-type seed. Dorsal view of a seed of *S. latifolia* with the EFT curve superimposed in green and the corresponding curvature values plotted. The EFT curve results from 98 points taken along the silhouette and 74 harmonics, following the protocol published [20]. The numbers indicate the correspondence between individual tubercles and their curvature values. The program that provides the EFT curve contains the algorithm to obtain and represent the curvature values along it (see Appendix A). Maximum curvature values correspond to individual tubercles. Bar represents 1 mm.

**Figure 8 plants-12-02439-f008:**
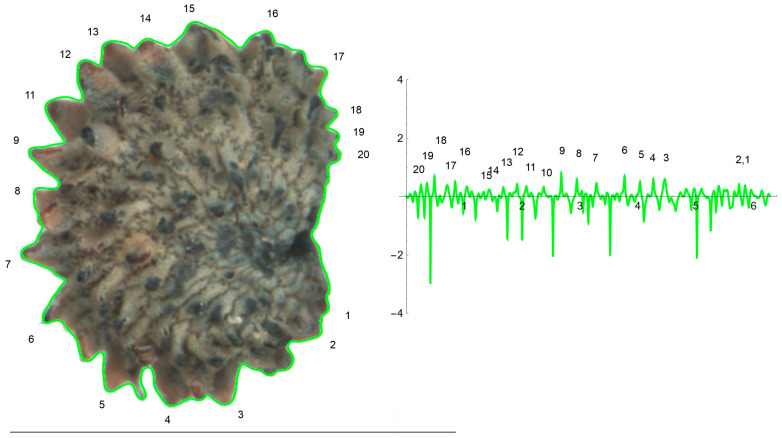
EFT and curvature analysis in a papillose-type seed. Lateral view of a seed of *S. perlmanni* with the EFT curve superimposed in green and the corresponding curvature values plotted. The EFT curve shown resulted from 189 points taken along the silhouette and 74 harmonics, following the protocol published [20]. The numbers indicate the correspondence between individual tubercles and their curvature values. The program that provides the EFT curve contains the algorithm to obtain and represent the curvature values along it (see Appendix A). This is a papillose-type seed, with tubercles of varied shape disposed unevenly at the seed surface. The maximum and minimum curvature values correspond to particular tubercles. Bar represents 1 mm.

**Figure 9 plants-12-02439-f009:**
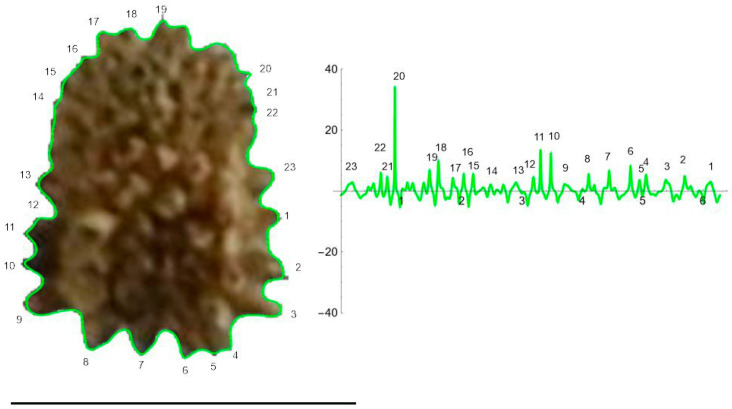
EFT and curvature analysis in a papillose-type seed. Dorsal view of a seed of *S. perlmanii* with the EFT curve superimposed in green and the corresponding curvature values plotted. The maximum and minimum curvature values correspond to individual tubercles. The EFT curve shown results from 184 points taken along the silhouette and 60 harmonics, following the protocol published [20]. The numbers indicate the correspondence between individual tubercles and their curvature values. Bar represents 1 mm. The program that provides the EFT curve contains the algorithm to obtain and represent the curvature values along it (see Appendix A). Bar represents 1 mm.

**Figure 10 plants-12-02439-f010:**
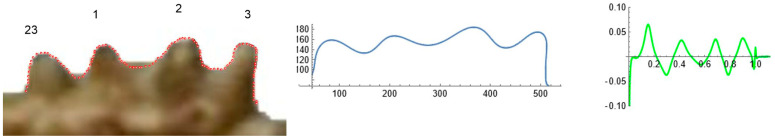
Curvature analysis of individual tubercles in *S. perlmanii.* Right: Four tubercles of *S. perlmanii*, numbers 23, 1, 2 and 3, from Figure 9 were selected to measure curvature individually. The corresponding Bézier curve (middle) and curvature plot (right) are shown in blue and green, respectively.

**Figure 11 plants-12-02439-f011:**
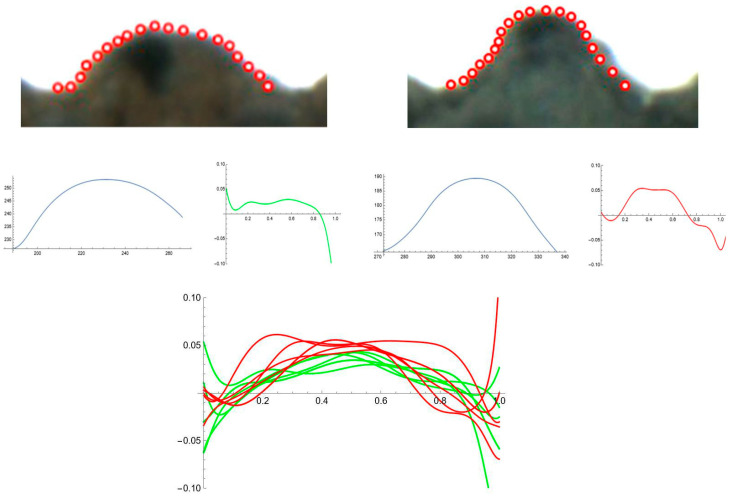
Curvature analysis for individual tubercles of *S. chlorifolia* and *S. latifolia.* Above: representative individual tubercles with points taken. Middle: curves and curvature plots. Below: plot of curvatures corresponding to five tubercles for each species. Green: *S. chlorifolia*; Red: *S. latifolia*.

**Figure 12 plants-12-02439-f012:**
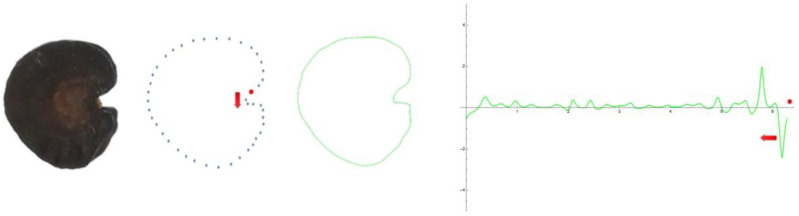
Example of the method (Method 1) to obtain EFT curve and curvature values from a seed image. Seed of *S. colorata* (lateral view), set of points selected, curve and results of curvature analysis. The red dot marks the initial point; the series of points follow clockwise, and the results of curvature analysis are shown counterclockwise, starting from the last point.

**Table 1 plants-12-02439-t001:** Morphological characteristics of the lateral and dorsal views of the seed populations subject of this work. A = area (mm^2^); P = perimeter (mm); L = length (mm); W = width (mm); AR = aspect ratio; C = circularity; R = roundness; S = solidity. Different superscript letters indicate significant differences between files for the measurement indicated. The coefficients of variation are indicated between parentheses.

	**Lateral View**							
**Species**	**A**	**P**	**L**	**W**	**AR**	**C**	**R**	**S**
*S. chlorifolia*	1.10 ^c^ (6.9)	4.32 ^b^ (3.4)	1.31 ^c^ (3.3)	1.07 ^c^ (4.9)	1.23 ^b^ (5.0)	0.74 ^c^ (3.9)	0.81 ^b^ (4.9)	0.96 ^b^ (0.7)
*S. colorata*	1.65 ^d^ (6.6)	5.27 ^c^ (3.2)	1.57 ^d^ (3.9)	1.34 ^d^ (3.1)	1.17 ^a^ (2.7)	0.75 ^c^ (1.8)	0.86 ^c^ (2.7)	0.96 ^b^ (0.6)
*S.latifolia*	1.01 ^b^ (2.0)	4.26 ^a b^ (2.6)	1.26 ^b^ (2.0)	1.03 ^b^ (1.7)	1.23 ^b^ (3.1)	0.70 ^b^ (4.2)	0.82 ^b^ (3.2)	0.96 ^b^ (0.4)
*S. perlmanii*	0.54 ^a^ (12.2)	4.07 ^a^ (11.0)	0.94 ^a^ (6.4)	0.73 ^a^ (6.6)	1.28 ^c^ (4.7)	0.42 ^a^ (16.3)	0.79 ^a^ (5.0)	0.88 ^a^ (2.5)
	**Dorsal View**							
**Species**	**A**	**P**	**L**	**W**	**AR**	**C**	**R**	**S**
*S. chlorifolia*	0.94 ^b^ (10.4)	4.08 ^b^ (5.0)	1.37 ^c^ (4.8)	0.88 ^c^ (6.8)	1.56 ^b^ (5.9)	0.71 ^c^ (3.5)	0.64 ^b^ (5.7)	0.96 ^c^ (1.4)
*S. colorata*	0.53 ^a^ (18.7)	4.03 ^b^ (6.1)	1.36 ^c^ (5.0)	0.50 ^a^ (16.9)	2.82 ^c^ (15.3)	0.41 ^a^ (13.1)	0.36 ^a^ (16.1)	0.81 ^a^ (4.0)
*S.latifolia*	0.89 ^b^ (3.2)	3.95 ^b^ (1.9)	1.29 ^b^ (2.2)	0.88 ^c^ (2.9)	1.46 ^ab^ (4.0)	0.71 ^c^ (3.9)	0.68 ^c^ (3.9)	0.96 ^c^ (0.6)
*S. perlmanii*	0.49 ^a^ (13.4)	3.52 ^a^ (9.1)	0.92 ^a^ (7.4)	0.68 ^b^ (7.0)	1.34 ^a^ (4.8)	0.50 ^b^ (11.6)	0.75 ^d^ (4.9)	0.90 ^b^ (1.7)

**Table 2 plants-12-02439-t002:** Summary of curvature results for the comparison between *S. chlorifolia* and *S. latifolia*. ANOVA for six individual tubercles of each species. Different letters in superscript indicate significant differences between species.

Species	*S. chlorifolia*	*S. latifolia*
Maximum curvature values in six tubercles (mean)	40.7 ^a^	55.5 ^b^
Mean curvature values in six tubercles (mean)	17.2 ^a^	24.5 ^b^

**Table 3 plants-12-02439-t003:** Summary of curvature results for the comparison between *S. chlorifolia* and *S. latifolia*. ANOVA for 18 individual tubercles corresponding to three seeds of each species. Different letters in superscript indicate significant differences between species. Different superscript letters indicate significant differences between files for the measurement indicated. The coefficients of variation are indicated between parentheses.

Species	*N*	*S. chlorifolia*	*S. latifolia*
Maximum curvature values	18	47.1 ^a^ (60.9)	97.8 ^b^ (42.7)
Mean curvature values	18	32.0 ^a^ (69.8)	61.3 ^b^ (38.5)

**Table 4 plants-12-02439-t004:** List of seed populations analyzed in this work. The populations JBUV 519, JBUV100 and JBUV 1444 were obtained from the carpoespermateca at the Botanical Garden of the University of Valencia and proceed from an exchange protocol between seed collections through the world.

Species	Source (Place of Origin)	Lifespan	Morphological Type [10,11]	Subgenus and Section [21]
*S. perlmanii* W.L.Wagner, D.R.Herbst and Sohmer	JBUV 519 * Botanischer Garten der Universität Zürich (U **)	Annual	Papillose	*S.* subg. *Silene* sect. Sclerophyllae (Chowdhuri) F.Jafari, Oxelman and Rabeler
*S. colorata* Poir. (AJ301)	Ana Juan (Villena, Alicante, Spain)	Annual	Smooth	*S.* subg. *Silene* sect. *Silene*
*S. chlorifolia* Sm.	JBUV100 BG der Martin-Luther-Univ. Halle-Wittenberg (U **)	Perennial	Rugose	*S.* subg. *Silene* sect. Sclerocalycinae (Boiss.) Schischk. in Komarov
*S. latifolia* Poir	JBUV 1444 * Humboldt University, Berlin (Germany, Brandenburg, Landkreis Märkisch-Oderland, Petershagen)	Perennial	Echinate	*S.* subg. *Behenantha* (Otth) Torr. and A.Gray sect. *Melandrium* (Röhl.) Rabeler

* JBUV = Jardín Botánico Universidad de Valencia. ** U stands for unknown.

## Data Availability

The data presented in this study are available in Appendix A.

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
