# Peer review of "Curvature Analysis of Seed Silhouettes in Silene L."

_plants, 2023, doi:10.3390/plants12132439_

Round 1

Reviewer 1 Report

It is an interesting and promising approach for automatic species identification by seed. The MS is clear and well written, but there's a substantive stumbling block: what about static support of the data? How many seeds from an accession was measured and analyzed? Didn't find any reference in the text. It appears that only a single seed from each of the four species was measured and analyzed.

Author Response

Dear Reviewer,

Thank you very much for your comments. Following your advice, the comparison between individual tubercles of S. chlorifolia and S. latifolia has been improved and consists now of two experiments. In the first experiment, curvature values were measured on six individual tubercles for each species. The points defining the individual tubercles were obtained manually and the results correspond to Figure 11.

New data have been added with the analysis of three seeds per species (six tubercles per individual seed, i.e. 18 tubercles per species). In this case, the points for the individual tubercles were obtained automatically with the function “Analyze line graph of Image J”, as indicated now in the Materials and Methods section. Corresponding details on the statistics (ANOVA) are also given in the materials and methods and results sections.

Reviewer 2 Report

The article is scientifically relevant and important for the scholars related to this topic. Though, there are some aspects that must be improved (please, see file attached):

i) Introduction, it is clear and congruent with the other sections of the article. However, the objective of the study is not clear enough; it can be inferred but it needs more forcefulness.

ii) Though the structure of the article is quite flexible, according to the research logic (experimental method), to first read the Material and Methods section, allowed the reader to a better understanding of what have been done, and to provide a better context and understanding of the Results obtained. In this logic, after the Results, there is the Discussion and, finally, the Conclusion.

In this case, after the Introduction, there is the Results, the Discussion, Material and Methods, and the Conclusion. Hence, it is not easy to question or understand the results without knowing what was done and why; there is a lack of context. Same feeling is got when the Conclusion comes after the Material and Methods sections.

iii) Material and Methods, there is not clear if six or seven or any other number of seeds, per species, were used for this study, in the sense of getting robust results (enough number of replicates, and objective and reasonable results). This study includes a statistical analysis, though, this part is not developed at all; some information is given in the Discussion, but it should be explained in this section.

About the seeds, authors mention that the seeds were provided by the spermatec; thus, it is important to know the physical state of the seeds (dry, fresh, how are they being kept, for how long, etc.), because when the seeds are dry, their shape and size diminished, the color may change, etc.

iv) Results, it is important to give a small description of the seeds, per species, and of their habitat for a better understanding of the natural differences among them.

In the figures, seed´s pictures lack context in terms of size, a bar could be very helpfully.

In addition, Tables and Figures need to be more explicative. They should be understandable without reading the whole article, that is why a title and a developed legend are so important (including any note: abbreviation meaning, statistical significant differences, etc.).

v) Discussion, congruent with the Results.

vi) Congruent with the Results and Discussion, and purposeful.

vii) References, congruent but need to standardize the style of presentation.

Author Response

Dear Reviewer,

Thank you very much for your commentaries that have contributed notably to improve the quality of the article.

The introduction has been modified to put more emphasis on the objectives of this work. The last paragraphs of the introduction emphasize the application of the described techniques to the analysis of seed shape in Silene species.

The layout of the sections is mandatory for this journal and the Materials and methods section is placed at the end. However, in our experience, readers usually read this section first to get an idea of the experiments carried out.

The Materials and Methods section has been strengthened with details on seed number and statistical analysis.

In the first part of the results, as well as in the Materials and methods section, new information on the seeds is given. At the beginning of the results, a new section includes the general morphological analysis. It gives an idea of the general condition of the seeds. Although they have been stored for varying periods of time, they keep their morphological characteristics relatively constant (low values for coefficient of variation; high values for robustness).

In addition, the information in the Materials and methods section concerning the seeds has been improved (Table 4).

A bar has been added for control of size in all the figures.

The legends to figures and tables have been expanded to be more descriptive. They contain a title and include explanations about statistical differences when appropriate.

All the corrections indicated in the PDF file have been included in the current version of the article.

Reviewer 3 Report

Seed shape is an important trait in plant identification and classification. The manuscript investigated the complexity of lateral and dorsal views of seeds of Silene species by the application of the Elliptic Fourier Transform (EFT). They demonstrated that the method can successfully discriminate between various seed groups. Curvature analysis revealed differences between representative seeds, with lower values in rugose seeds, and higher in tubercles of the echinate seeds.

Some minor comments:

Elliptic Fourier Transform has been used for seed shape analysis in other research. Is there any improvement in your research? And the advantage of EFT compared with other seed morphology methods.

Author Response

Dear Reviewer,

Thank you very much for your commentaries.

It is true that the Elliptic Fourier Transform has already had many applications in the field of morphology and for the analysis and comparison of seed shape.

In our case, this is the second article in which we have used this protocol. In the first case (reference 20 of this article), EFT was applied to Silene seeds to obtain models for these seeds that do not fit very well to known geometrical figures. The program we designed to generate an EFT curve from a series of points also contains the code for the calculation of the curvature values along the curve and its corresponding plotting. The main purpose of this article was to obtain the curvature values by various methods, so the focus was not on the EFT curves themselves, but on the analysis of the curvature along them.

Round 2

Reviewer 1 Report

It is OK now.

Reviewer 2 Report

Authors have done all the correcions suggested; although, two small corrections have to be done:

Line 79, Silene in Italics.

Line 428, popilations, ANOVA. Add a ","

The article has improved its quality.